# Two-Dimensional Nitronyl Nitroxide–Cu Networks Based on Multi-Dentate Nitronyl Nitroxides: Structures and Magnetic Properties

**Hongdao Li [1,2,*], Jiao Lu [2], Jing Xie [2], Pei Jing [2] and Licun Li [2,*]**

[1] Department of Chemistry and Chemical Engineering, Taiyuan Institute of Technology, Taiyuan 030008, China
[2] Department of Chemistry, Key Laboratory of Advanced Energy Materials Chemistry, College of Chemistry, Nankai University, Tianjin 300071, China; lj96111@126.com (J.L.); 13662036932@163.com (J.X.); jingpei22@163.com (P.J.)
[*] Correspondence: lihong.dao@163.com (H.L.); llicun@nankai.edu.cn (L.L.)

**Abstract:** Two multi-dentate nitronyl nitroxide radicals, namely, bisNITPhPy ([5-(4-pyridyl)-1,3-bis(1′-oxyl-3′-oxido-4′,4′,5′,5′-tetramethyl-4,5-hydro-1*H*-imidazol-2-yl)]benzene) and NIT-3Py-5-4Py (2-{3-[5-(4-pyridyl)]pyridyl}-4,4,5,5-tetramethylimidazoline-1-oxyl-3-oxide), were assembled with $Cu^{II}$ ions to obtain two-dimensional heterospin 2p–3d coordination polymers $[Cu_7(hfac)_{14}(bisNITPhPy)_2]_n$ (**1**) and $[Cu_2(hfac)_4(NIT-3Py-5-4Py)]_n$ (**2**) (hfac: hexafluoroacetylacetonate). In both compounds, the bisNITPhPy and NIT-3Py-5-3Py radicals acted as pentadentate and tetradentate ligands, respectively, to connect with $Cu^{II}$ ions, generating a 2D layer structure. The analysis of the magnetic behavior indicated that strong antiferromagnetic coupling and ferromagnetic interaction ($J$ = 17.1 cm$^{-1}$) coexisted in **1**. For **2**, there were ferromagnetic couplings between the $Cu^{II}$ ion and NO group, as well as the $Cu^{II}$ ion and radical via the pyridine ring with $J_1$ = 32.8 and $J_2$ = 2.2 cm$^{-1}$, respectively.

**Keywords:** multi-dentate nitronyl nitroxide; Cu(II) ion; two-dimension; magnetic properties

## 1. Introduction

The metal–radical heterospin strategy, in which metal ions link with stable organic radicals, is extremely fascinating for designing molecular magnetic materials [1–6]. Based on this kind of heterospin approach, 3*d* [7–9], 4*f* [10–17], and 3*d*–4*f* [18–23] compounds have been obtained so far by employing various organic radicals, such as $N_2^{3-}$ [15,16], $HAN^{3-}$ [24], $TTF^+$ [25], thiadiazoyl [14], and nitronyl nitroxide radical ligands [6,7,10,13,19,23,26–32]. In particular, nitronyl nitroxide radicals are often used as building blocks to construct metal compounds with various topology structures due to the substituent groups of nitronyl nitroxides being able to effectively regulate the spatial arrangement of magnetic building blocks [33]. However, it is worth noting that most of the nitronyl nitroxide–metal compounds display zero- [34–37] and one-dimensional (1D) structures [10,11,26,27], while nitronyl nitroxide radicals bridged two-dimensional (2D) metal complexes are scarce so far. This could be attributed to the fact that metal centers require electron-attracting coligands, such as hexafluoroacetylacetonate, to promote the coordination of nitronyl nitroxide radicals. Nevertheless, these coligands occupy some coordination sites of the metal center and possess a large steric hindrance, going against the formation of higher-dimensional structures. In this regard, multi-dentate nitronyl nitroxide with multiple coordination groups, such as functionalized biradicals and mono-radicals with two additional coordination groups, is able to link several metal ions to generate high-dimensional heterospin complexes. Recent examples of 2D compounds involving multi-dentate nitronyl nitroxide illustrate this point [38–41].

Along this line, to further expand nitronyl-nitroxide-based 2D heterospin systems, herein we utilized functionalized biradical bisNITPhPy ([5-(4-pyridyl)-1,3-bis(1′-oxyl-3′-

oxido-4′,4′,5′,5′-tetramethyl-4,5-hydro-1*H*-imidazol-2-yl)]benzene) and tetradentate mono-radical NIT-3Py-5-4Py (2-{3-[5-(4-pyridyl)]pyridyl}-4,4,5,5-tetramethylimidazoline-1-oxyl-3-oxide) (Scheme 1) to bridge Cu ions, constructing two novel 2D rad–Cu complexes, namely, $[Cu_7(hfac)_{14}(bisNITPhPy)_2]_n$ (**1**) and $[Cu_2(hfac)_4(NIT-3Py-5-4Py)]_n$ (**2**). Magnetic property investigations were undertaken, where ferromagnetic and strong antiferromagnetic Cu(II)–rad exchanges were observed in compound **1,** while ferromagnetic coupling dominated the magnetic system of complex **2**.

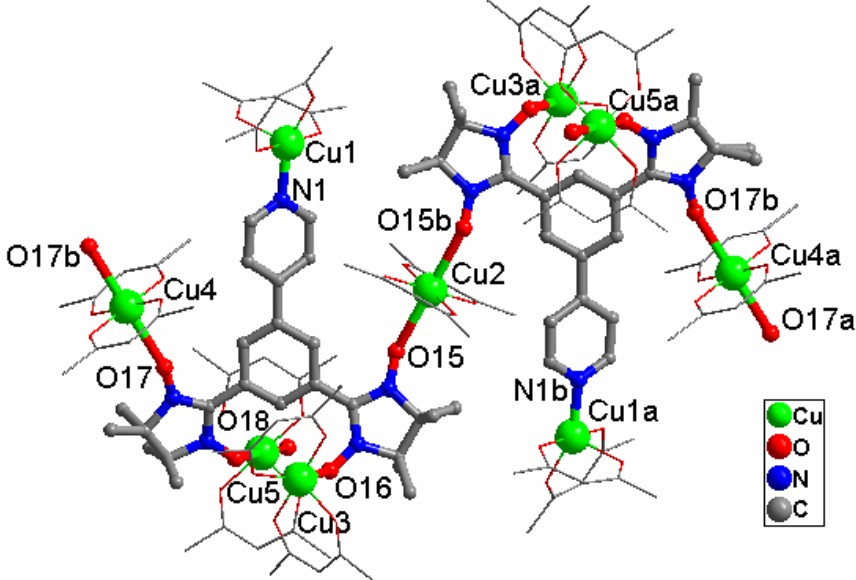

**bisNITPhPy**          **NIT-3Py-5-4Py**

**Scheme 1.** bisNITPhPy and NIT-3Py-5-4Py ligands.

## 2. Results and Discussion

### 2.1. Structural Investigation and Description

X-ray diffraction of both Cu$^{II}$ compounds **1** and **2** showed that they all crystallized in the triclinic space group *P*ī. The particulars of the crystallographic structural data of two-dimensional polymers are included in Table 1. For compound **1**, the asymmetric part contained seven Cu(hfac)$_2$ moieties and two bisNITPhPy biradical ligands (Figure 1). In **1**, one NIT unit of each bisNITPhPy ligand bridged two different copper(II) ions (Cu4 and Cu5) through its two NO groups, leading to infinite linear chains. Meanwhile, 1D chains were connected to Cu(II) ions (Cu2 and Cu3) via another NIT moiety with the [Cu3-NIT-Cu2-NIT-Cu3] structural units to generate one two-dimensional layer. Meanwhile, the N atom of the pyridine ring of each bisNITPhPy ligand was connected to one Cu$^{II}$ ion (Cu1) in its equatorial position (Figure 2).

**Figure 1.** Asymmetric unit of the reticular complex in **1**. The H and F atoms are left out (a: −x + 2, −y, −z; b: −x + 2, −y + 1, −z).

**Table 1.** Crystallographic structural data of two-dimensional polymers **1** and **2**.

| Complex | 1 | 2 |
|---|---|---|
| Formula | $C_{120}H_{76}Cu_7F_{84}N_{10}O_{36}$ | $C_{53}H_{38}F_{36}Tb_2N_6O_{17}$ |
| $M$ (g·mol$^{-1}$) | 4274.76 | 1266.67 |
| $T$ (K) | 113(2) | 113(2) |
| Crystal system | Triclinic | Triclinic |
| Space group | $P\bar{1}$ | $P\bar{1}$ |
| $a$ (Å) | 13.4647(13) | 11.172(2) |
| $b$ (Å) | 16.7473(18) | 15.205(3) |
| $c$ (Å) | 18.793(2) | 15.890(3) |
| $\alpha$ (°) | 109.221(2) | 78.94(3) |
| $\beta$ (°) | 96.022(2) | 85.98(3) |
| $\gamma$ (°) | 98.901(2) | 70.21(3) |
| $Z$ | 1 | 2 |
| $D_{calcd}$ (g·cm$^{-3}$) | 1.821 | 1.688 |
| $\mu$ (mm$^{-1}$) | 1.113 | 0.998 |
| $\theta$ (°) | 3.00–25.00 | 1.93–25.00 |
| $F(000)$ | 2113 | 1254 |
| Reflns collected | 36,087 | 24,124 |
| Unique reflns/$R_{int}$ | 13,715/0.0429 | 8784/0.0598 |
| GOF ($F^2$) | 0.983 | 1.007 |
| $R_1/wR_2$ ($I > 2\sigma(I)$) | 0.0497, 0.1412 | 0.0927, 0.2159 |
| $R_1/wR_2$ (all data) | 0.0663, 0.1511 | 0.1149, 0.2363 |

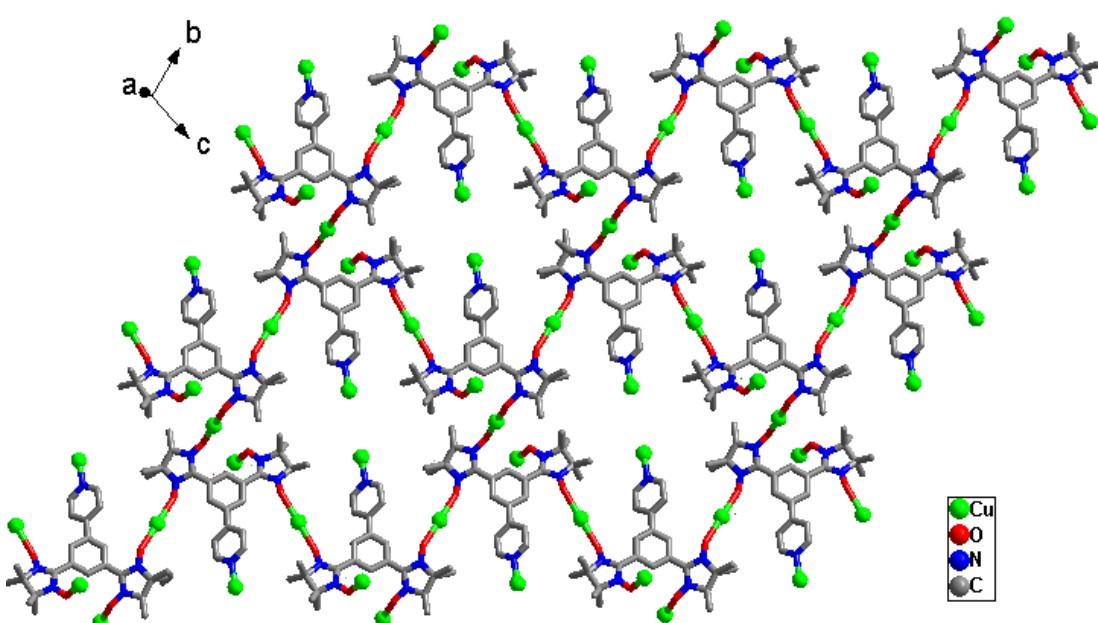

**Figure 2.** Network structure of compound **1**. The H atoms and bidentate hfac$^-$ ligands are deleted.

Cu2, Cu4, and Cu5 adopted distorted octahedral geometries (Figure S1). The equatorial positions were filled by two ß-diketonate coligands (1.935(2)–1.951(2) Å for Cu2–O$_{hfac}$, 1.930(3)–1.943(3) Å for Cu4–O$_{hfac}$ and 1.941(3)–1.950(3) Å for Cu5–O$_{hfac}$), while the apical sites were held by two NO groups from two mono-radicals (Cu(2)–O(15): 2.420(1) Å, Cu(4)–O(17): 2.428(3) Å, and Cu(5)–O(18): 2.330(1) Å). These axial bond lengths were longer than equatorial distances, suggesting the Jahn–Teller effect was in action [42,43]. Five-coordinated Cu1 and Cu3 both had distorted pyramidal geometries, which were assessed using a shape program [44] (Figure S1, Table S3). The coordination environment of Cu1 was built of a pyridyl N atom of the radical and four O atoms of the hfac$^-$ ligands, in which the O3 atom was located in the axial position (Cu1–O3: 2.166(3) Å). For

Cu3, the equator plane was generated by several O atoms (O16, O5, O7, and O8) from a nitroxide unit of the bisNITPhPy ligand (Cu3–O16: 1.989(3) Å) and two hfac$^-$ ligands (Cu–O: 1.904(3)–1.963(3) Å), and the O6 atom of one hfac$^-$ ligand occupied the vertex (Cu3–O6: 2.146(3) Å). Within the two-dimensional sheet, the shortest distance of Cu$\cdots$Cu was 6.246 Å (Figure 2). Adjacent 2D planes stacked with the closest interlayer Cu$\cdots$Cu distance of 9.364 Å.

For [Cu$_2$(hfac)$_4$(NIT-3Py-5-4Py)]$_n$ (**2**), asymmetric elements were made up of two Cu(hfac)$_2$ units and one NIT-3Py-5-4Py radical ligand (Figure 3). As shown in Figure 4, two NIT-3Py-5-4Py radicals linked Cu$^{II}$ ions by relying on nitroxide groups and nitrogen atoms of pyridines to produce an annular structure [Cu$_2$(NIT-3Py-5-4Py)$_2$]. Meanwhile, each dimer ring was linked to two Cu(hfac)$_2$ units via two nitroxide units, yielding a 1D chain. Meanwhile, adjacent 1D chains coordinated with Cu$^{II}$ ions via N atoms of the pyridine rings to construct a network structure.

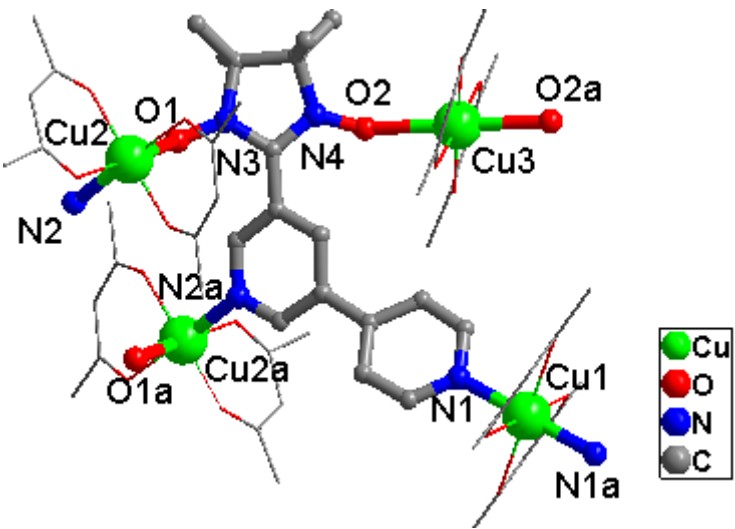

**Figure 3.** Asymmetric unit of the reticular complex in **2**, in which the H and F atoms are left out (a: −x, −y + 2, −z − 1).

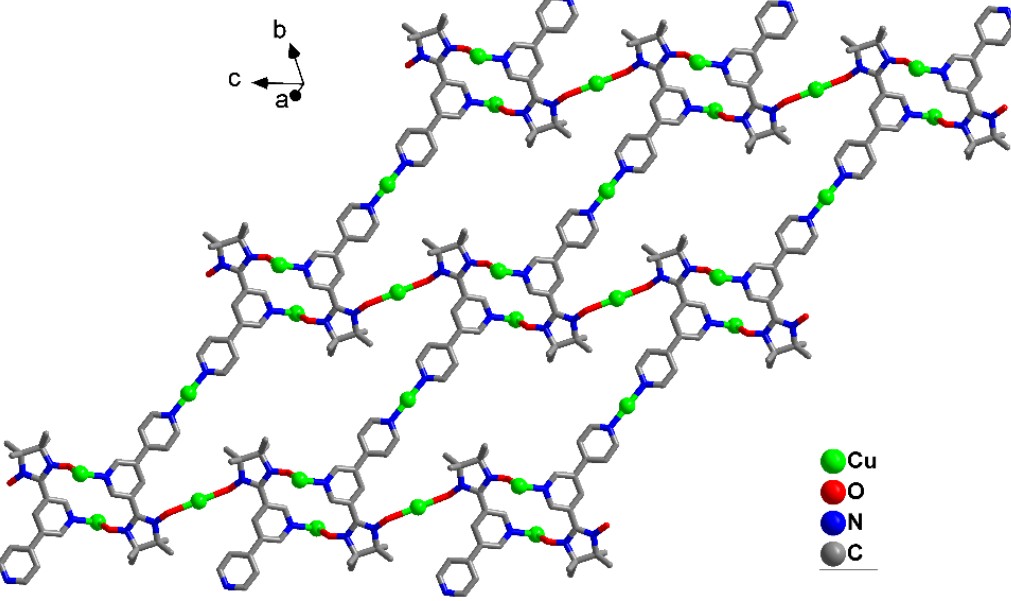

**Figure 4.** The 2D layer structure of complex **2**. In order to clearly represent the structure, H atoms and the bidentate hfac$^-$ are left out.

These Cu$^{II}$ ions all adopted an elongated octahedral geometry (Figure S3). For Cu1, the apical positions were taken over by two oxygen atoms that arose from two different hfac$^-$ ligands (average Cu–O: 2.230(6) Å), while the equatorial plane took shape via two nitrogen atoms from the two pyridine rings of two radicals (average Cu–N bond: 2.017(4) Å) and two oxygen atoms belonging to hfac$^-$ ligands (average Cu–O bond: 2.039(5) Å). For Cu3, two O atoms derived from two NO units lay in axial sites (the average Cu–O$_{rad}$: 2.362(5) Å). In addition, four O atoms belonging to different hfac$^-$ ligands produced a geometric equatorial plane (Cu–O$_{hfac}$: 2.229(5)–1.947(5) Å). For Cu2, one nitrogen atom of one pyridine ring and three oxygen atoms of two hfac$^-$ ligands occupy equatorial sites (Cu-N: 2.061(6) Å, Cu–O$_{hfac}$: 1.945(6)–2.002(6) Å). Meanwhile, an oxygen atom originating from one nitroxide group of the radical ligand and the other being from one hfac$^-$ ligand were located in axial positions (Cu–O$_{rad}$: 2.326(5) Å and Cu–O$_{hfac}$: 2.237(6) Å). In a 2D layer, the nearest Cu··· Cu separation was 5.911 Å (Figure 4). The shortest interlayer Cu··· Cu interval between adjacent networks was 11.172 Å.

### 2.2. Magnetic Properties

A direct current magnetic susceptibility study of [Cu$_7$(hfac)$_{14}$(bisNITPhPy)$_2$]$_n$ (**1**) was implemented over the 2–300 K range with an extrinsic dc field of 5000 Oe. As described in Figure 5, the $\chi_M T$ value of [Cu$_7$(hfac)$_{14}$(bisNITPhPy)$_2$]$_n$ (**1**) at 300 K was 3.15 cm$^3$Kmol$^{-1}$, which was lower than the desired value of 4.125 cm$^3$Kmol$^{-1}$ for uncoupled seven copper ions and four mono-radicals, but close to the theoretical values of 2.625 cm$^3$Kmol$^{-1}$ for uncoupled five copper ions and two mono-radicals (Cu$^{II}$: $C = 0.375$ cm$^3$Kmol$^{-1}$ and $S = 1/2$; meanwhile, for the radical: $S = 1/2$), suggesting strong antiferromagnetic coupling. On cooling, the $\chi_M T$ value increased slowly during the range of 300–35 K and then further increased abruptly to reach the maximum value of 8.36 cm$^3$Kmol$^{-1}$ at 2.0 K. The above phenomenon shows that strong antiferromagnetic coupling and ferromagnetic interaction coexisted in **1**.

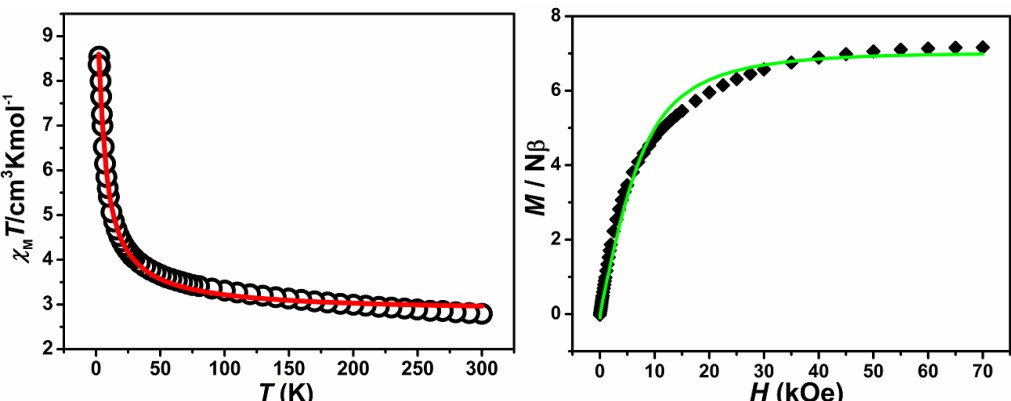

**Figure 5.** (**left**) $\chi_M T$ versus $T$ chart of [Cu$_2$(hfac)$_4$(NIT-3Py-5-4Py)]$_n$ (**2**) (the crimson line denotes the fitting results). (**right**) *M* versus *H* chart for **1** at 2 K (the green curve denotes the calculated behavior for the total of the Brillouin functions with seven free $S = 1/2$ spins).

Based on the structure of compound **1**, the O16 atom of the NO unit lay on one side of the Cu3 equatorial plane to generate strong antiferromagnetic interaction and completely offset the opposite spin [45,46]. Furthermore, Cu$^{II}$–rad magnetic coupling via the pyridyl and phenyl rings might be very weak [47]. Thus, the magnetic behavior of **1** was mainly derived from where the mono-radical bridged two Cu$^{II}$ chains involving Cu4 and Cu5 ions and uncoupled three copper ions (Cu1, Cu2, and Cu1a) (Scheme 2). The observed magnetic behavior of **1** should be analyzed using Equations (1)–(4). For the Cu$^{II}$ chain,

an equation (Equation (2)) for the magnetic susceptibility was based on the Hamiltonian $\hat{H} = -J\hat{S}_{rad} \cdot \hat{S}_{Cu}$ [48]. The mean field ($zJ'$) denotes additional magnetic coupling.

$$\chi = 4\chi_{rad-Cu(chain)} + 3\chi_{Cu} \tag{1}$$

$$\chi_{rad-Cu(chain)} = \frac{Ng^2\beta^2}{4kT} \times \left[\frac{N}{D}\right]^{\frac{2}{3}} \tag{2}$$

$$N = 1.0 + 5.7979916y + 16.90253y^2 + 29.376885y^3 + 29.832959y^4 + 14.036918y^5$$

$$D = 1.0 + 2.7979916y + 7.0086780y^2 + 8.6538644y^3 + 4.5743114y^4$$

$$y = \frac{J}{2kT}$$

$$\chi_{Cu} = \frac{Ng^2\beta^2}{3kT}\left[\frac{1}{2}\left(\frac{1}{2}+1\right)\right] \tag{3}$$

$$\chi_{total} = \frac{\chi}{1 - \left(\frac{2zJ'\chi}{Ng^2\beta^2}\right)} \tag{4}$$

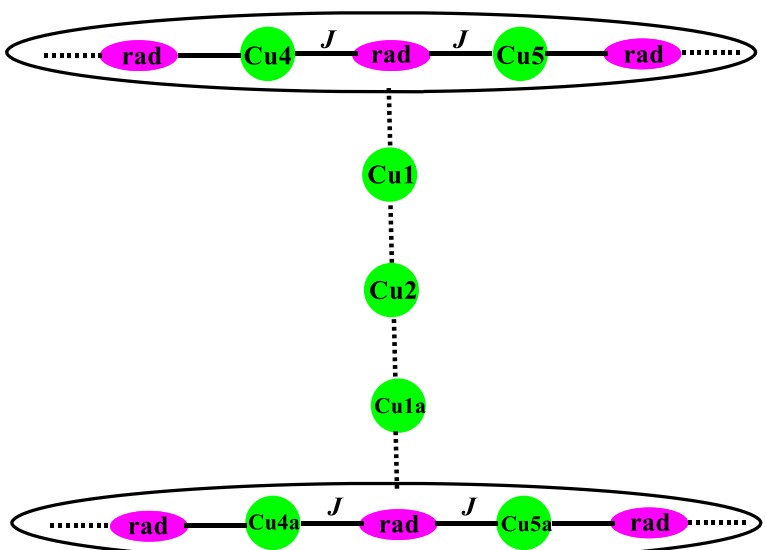

**Scheme 2.** Magnetic exchange model in **1**.

The rational evaluation of the measurement data produced $J$ = 17.11 cm$^{-1}$, $g$ = 2.08, $zJ'$ = −0.044 cm$^{-1}$, and $R^2$ = 0.9963. The positive $J$ parameter signifies the ferromagnetic Cu–radical coupling, which was derived from the axial Cu$^{II}$–ON bond to give rise to orbital orthogonality of the $\pi^*$ orbital of the NO group and the $d_{x^2-y^2}$ orbital of the Cu$^{II}$ ion [49,50]. The magnetic coupling is analogous to some reported Cu$^{II}$–nitronyl nitroxide compounds [42,51].

The isothermal $M$ versus $H$ curve for complex **1** was investigated at 2.0 K in 0–70 kOe (Figure 5). The magnetization displayed a sharp rise below 15 kOe, suggesting the existence of ferromagnetic couplings, in agreement with the precipitous increase of the $\chi_M T$ value in the low-temperature range. Then, $M$ increased slowly to 7.16 Nβ at 70 kOe, in accordance with the theoretical values (7.0 Nβ) found using the Brillouin functions for free seven $S$ = 1/2 spins, which verified the strong Cu$^{II}$–rad antiferromagnetic interaction in the equatorial plane.

The $\chi_M T$–$T$ curve for [Cu$_2$(hfac)$_4$(NIT-3Py-5-4Py)]$_n$ (**2**) is described in Figure 6. The room temperature value of $\chi_M T$ was 1.21 cm$^3$Kmol$^{-1}$, which approached the desired value of 1.125 cm$^3$Kmol$^{-1}$ for two uncoupled Cu$^{II}$ ions and one radical. On cooling, the

$\chi_M T$ value of compound **2** increased steadily to 4.66 cm$^3$Kmol$^{-1}$ at 2 K, showing that ferromagnetic coupling dominated the 2*p*–3*d* magnetic system.

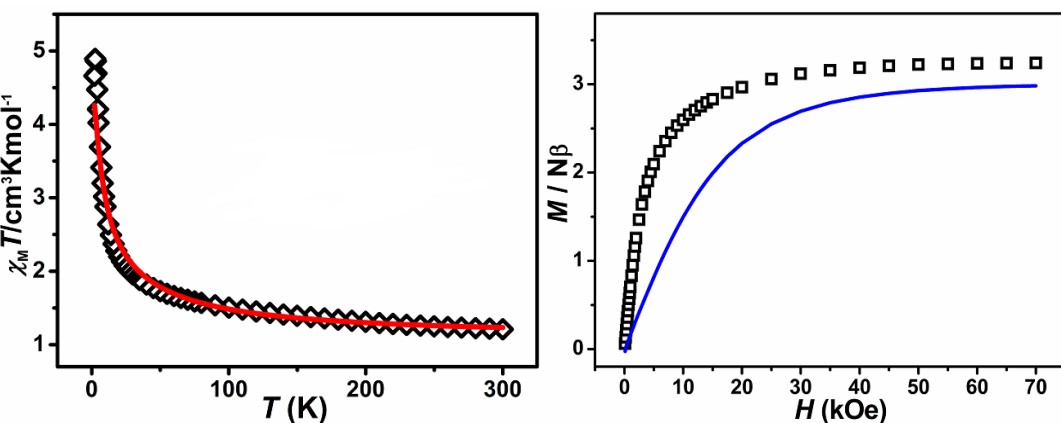

**Figure 6.** (**left**) $\chi_M T$ versus *T* chart of the reticular complex in **2** (the crimson curve denotes the fitting result). (**right**) *M* versus *H* chart at 2 K (the blue curve denotes the sum of the Brillouin function).

For the magnetic system of compound **2**, a negligible magnetic Cu$^{II}$–NIT exchange through the two pyridyl rings was anticipated. Based on the above analysis, two effective exchange pathways should be considered: (i) the directly axial coordinated NO–Cu exchange ($J_1$) and (ii) the Cu–NO coupling via pyridine heterocycle ($J_2$). Accordingly, from a magnetic point of view, this 2D complex could be considered as the radicals bridged 1D chains plus the uncoupled mononuclear copper(II) units. There is no available magnetic expression for such a 1D system. Thus, the MAGPACK procedure was employed to investigate the dc magnetic susceptibilities involving a closed cycle containing two [Cu-{Cu$_2$NIT$_2$}] units (Scheme 3) with two additional uncoupled Cu$^{II}$ ions with the Hamiltonian.

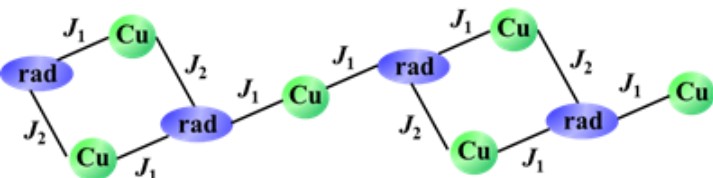

**Scheme 3.** Magnetic coupling model in **2**.

$$\hat{H} = -2J_1 \left( S_{rad_1} S_{Cu_1} + S_{rad_2} S_{Cu_2} + S_{rad_2} S_{Cu_3} + S_{rad_3} S_{Cu_3} + S_{rad_3} S_{Cu_4} + S_{rad_4} S_{Cu_5} \right)$$
$$-2J_2 \left( S_{rad_1} S_{Cu_2} + S_{rad_2} S_{Cu_1} + S_{rad_3} S_{Cu_5} + S_{rad_4} S_{Cu_4} \right)$$

The best-fitting result gave $J_1$ = 32.8 cm$^{-1}$, $J_2$ = 2.2 cm$^{-1}$, and *g* = 2.16. The ferromagnetic interaction ($J_1$) as a result of the NO group was axially bound to the Cu$^{II}$ ion [49,50]. Meanwhile, the ferromagnetic Cu$^{II}$–rad coupling through one pyridine ring ($J_2$) was explained in terms of a spin-polarization model (Scheme S1), which is comparable to other Cu–rad compounds [19,52].

The magnetization value was 3.24 Nβ at 2 K for 70 kOe, which is in agreement with the desired result of 3 Nβ. Beyond that, the experimental magnetization (*M*) was larger than the theoretical curve involving the Brillouin functions with one *S* = 1/2 and two *S* = 1/2 uncoupled spins, indicating the existence of a ferromagnetic property.

## 3. Experimental Section

### 3.1. Raw Materials and Physical Investigation

Multi-dentate radicals bisNITPhPy [31] and NIT-3Py-5-4Py [53] were prepared according to the related literature. Elemental analyses of both 2D Cu complexes were

implemented by means of Perkin-Elmer elemental analytical equipment. A Bruker TENOR 27 spectrograph was used to measure the Fourier infrared spectra. The magnetic data of both compounds were corrected using Quantum Design SQUID VS and diamagnetic corrections of both Cu compounds were carried out using Pascal's constants.

### 3.2. Preparation of $[Cu_7(hfac)_{14}(bisNITPhPy)_2]_n$ (**1**)

Anhydrous $Cu(hfac)_2$ (0.0148 g, 0.03 mmol) was added to 25 mL of *n*-hexane with stirring and reflux for 2 h. Then, $CHCl_3$ (5 mL) with bisNITPhPy (0.0046 g, 0.01 mmol) was introduced into the above organic solution. After stirring for 30 min, a green solution was filtered and evaporated slowly to yield bottle-green crystals. Yield 75%. Anal. calc. for $C_{120}H_{76}Cu_7F_{84}N_{10}O_{36}$ (%): C 33.70, H 1.77, N 3.27; found. C 33.74, H 1.69, N 3.56; FT-IR (KBr, cm$^{-1}$): 3415 (s), 1651 (m), 1506 (m), 1366 (s), 1257 (s), 1209 (s), 1148 (s), 1134 (s), 947 (s), 861 (s), 663 (m), 587 (m), 546 (m).

### 3.3. Preparation of $[Cu_2(hfac)_4(NIT-3Py-5-4Py)]_n$ (**2**)

Anhydrous $Cu(hfac)_2$ (0.0097 g, 0.02 mmol) in 16 mL of *n*-hexane was refluxed for 3 h, into which NIT-3Py-5-4Py (0.0031 g, 0.2 mmol) in 7 mL of $CH_2Cl_2$ was introduced. The cyan solution was refluxed for about 0.5 h. The cyan solution was filtered and left for about 15 h to give cyan crystals. Yield 55%. Anal. calc. for $C_{37}H_{23}Cu_2F_{24}N_4O_{10}$ (%): C 35.09, H 1.83, N 4.42; found. C 35.21, H 2.28, N 4.40; FT-IR (KBr, cm$^{-1}$): 1650 (s), 1529 (m), 1465 (s), 1363 (m), 1251 (s), 1197 (s), 1133 (s), 799 (s), 662 (s), 578 (s), 526 (m).

### 3.4. Crystallographic Analysis

Structural data of both Cu$^{II}$ networks were gathered via a Rigaku Saturn diffractometer (Mo-K$\alpha$ source). Utilizing SADABS, the empirical absorption correction of two Cu$^{II}$ compounds was carried out. The structures of both compounds were parsed via direct methods and reasonably refined via least-squares with SHELXL-2014 on $F^2$ [54,55]. All H atoms were attached to suitable positions. For **2**, there was a large cavity of disordered solvent units, which was evaluated using the SQUEEZE [56] option. Anisotropic parameters were applied to all non-H atoms.

## 4. Conclusions

In summary, two novel rad–Cu heterospin 2D networks with different spin topologies were acquired by using multi-dentate nitronyl nitroxides. The magnetic behavior of compound **1** was mainly derived from mono-radical bridged Cu$^{II}$ 1D chains and [Cu-NIT-Cu-NIT-Cu] structural units. For compound **2**, the ferromagnetic interaction dominated the magnetic system, originating from 1D loop chains and uncoupled Cu$^{II}$ ions. Our work not only provides intriguing 2D nitronyl nitroxide-Cu compounds but also promises a new strategy for constructing two-dimensional magnetic materials through polydentate nitronyl nitroxide. This work shows that multi-dentate nitronyl nitroxide can act as fine linkers for facilitating an improvement in dimensionality to obtain a high-dimensional heterospin complex with fascinating magnetic properties.

**Supplementary Materials:** The following are available online at https://www.mdpi.com/article/10.3390/magnetochemistry7050073/s1. Tables S1 and S2: Vital bond lengths and corresponding angles for two Cu complexes; Tables S3 and S4: Structural analysis for Cu$^{II}$ ions in two compounds; Figures S1 and S3: The coordination polyhedron of Cu(II) ions in both Cu$^{II}$ compounds; Figures S2 and S4: Molecular packing arrangement for both Cu$^{II}$ complexes; Scheme S1: Spin polarization model of the Cu$^{II}$–rad interaction in compound **2**; CCDC 2079603 and 2079604 involve crystallographic data of Cu compounds, where these data can be downloaded from www.ccdc.cam.ac.uk/data_request/cif (accessed on 23 April 2021).

**Author Contributions:** L.L. produced the design for the experiment. H.L. and J.L. undertook the experiments and wrote and revised the article. H.L., J.L., J.X., P.J., and L.L. processed the structural and magnetic data. All authors have read and agreed to the published version of the manuscript.

**Funding:** The research was subsidized by the National Natural Science Foundation of China (No. 21773122).

**Institutional Review Board Statement:** Not applicable.

**Informed Consent Statement:** Not applicable.

**Data Availability Statement:** Not applicable.

**Conflicts of Interest:** The authors declare no conflict of interest.

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
