# Peer review of "Two-Dimensional Nitronyl Nitroxide–Cu Networks Based on Multi-Dentate Nitronyl Nitroxides: Structures and Magnetic Properties"

_magnetochemistry, doi:10.3390/magnetochemistry7050073_

Round 1

Reviewer 1 Report

The authors describe the synthesis, the structure, and the magnetic properties of two 2D Cu(II)-nitronyl nitroxide coordination polymers. The work is interesting and deals with infrequent spin carriers and interactions, but the following issues need to be resolved before publishing the work.

  1. Not everybody is familiar with these nitronyl nitroxide radicals, and it is not easy to figure out their structure just with their nomenclature. It would be good to have a scheme of both radicals in the Introduction section or at least in the supplementary material.
  2. The quality of Figure 1 and Figure 2 should be improved. The pixel size is too big; the dpi has to be increased.
  3. The Figure 5 caption should be corrected

the green line represents the theoretical “cure” for the sum of the

  1. page 5, line 123, the authors say “the corresponding maximum” I do not understand to whom correspondent is. I would say “to reach a maximum value of 8.36....”
  2. Page 5, last paragraph. According to the magnetic interactions between the radicals and the Cu(II) ions, the authors reduce the number of spin carriers from 11 to 7 and the dimensionality of the magnetic structure to “two mono-radical bridged CuII chains and uncoupled three copper ions.” This has to be illustrated with a scheme, at least, in the supplementary material. Which Cu(II) atoms involve the chains, Cu4 and Cu5?? Which are uncoupled? The authors should describe this aspect more precisely.
  3. Page 6 equation 3 is wrong, and there is no reference for eq (2)
  4. The fit looks good, but there is no “R” fitting parameter. Please provide
  5. In the description of the magnetization, the authors say, “with the Brillouin functions for two non-coupled S = 1/2 and five S = 1/2 spins, “ Previously, we had 4 spin carriers involved in the chains and 3 uncoupled. They converted into two non-coupled and five S =1/2. Please explain or correct.
  6. Figure 6 caption, please correct “cure” to curve.
  7. The magnetic chain is converted into a closed ring+ uncoupled Cu(II) ions. This is OK, but, How many uncoupled Cu(II) ions per 10 Cu(II) ions involved in the closed ring? A better description of the magnetic model is required.
  8. Experimental. It is always a good idea to include comparing the simulated and experimental powder diffraction patterns to ensure the integrity of the polycrystalline sample.

Author Response

Reviewer 1:

The authors describe the synthesis, the structure, and the magnetic properties of two 2D Cu(II)-nitronyl nitroxide coordination polymers. The work is interesting and deals with infrequent spin carriers and interactions, but the following issues need to be resolved before publishing the work.

  1. Not everybody is familiar with these nitronyl nitroxide radicals, and it is not easy to figure out their structure just with their nomenclature. It would be good to have a scheme of both radicals in the Introduction section or at least in the supplementary material.

Authors: According to referee’s suggestion, we have added a structural scheme of two radicals in the introduction section, please see the text.

  1. The quality of Figure 1 and Figure 2 should be improved. The pixel size is too big; the dpi has to be increased.

Authors: According to referee’s suggestion, we have improved the quality of Figure 1 and Figure 2, please see the main text.

  1. The Figure 5 caption should be corrected “the green line represents the theoretical “cure” for the sum of the…”

Authors: Thank you for your suggestion. We have corrected them.

  1. Page 5, line 123, the authors say “the corresponding maximum” I do not understand to whom correspondent is. I would say “to reach a maximum value of 8.36....”

Authors: According to referee’s suggestion, we have corrected the inappropriate expression, please see the main text.

  1. Page 5, last paragraph. According to the magnetic interactions between the radicals and the Cu(II) ions, the authors reduce the number of spin carriers from 11 to 7 and the dimensionality of the magnetic structure to “two mono-radical bridged CuII chains and uncoupled three copper ions.” This has to be illustrated with a scheme, at least, in the supplementary material. Which Cu(II) atoms involve the chains, Cu4 and Cu5?? Which are uncoupled? The authors should describe this aspect more precisely.

Authors: Thank you for your suggestion. According to referee’s suggestion, we have indicated magnetic exchange pathways for compound 1 with a scheme and added the description. Please see the main text.

  1. Page 6 equation 3 is wrong, and there is no reference for eq (2)

Authors: Sorry, that’s our carelessness. We have corrected the equation 3.

  1. The fit looks good, but there is no “R” fitting parameter. Please provide

Authors: According to referee’s suggestion, we have added the “R” fitting parameter.

  1. In the description of the magnetization, the authors say, “with the Brillouin functions for two non-coupled = 1/2 and five = 1/2 spins, “ Previously, we had 4 spin carriers involved in the chains and 3 uncoupled. They converted into two non-coupled and five S =1/2. Please explain or correct.

Authors: Yes, the Brillouin function should be calculated by seven non-coupled S = 1/2 spins.

  1. Figure 6 caption, please correct “cure” to curve.

Authors: We have corrected the spelling mistake.

  1. The magnetic chain is converted into a closed ring+ uncoupled Cu(II) ions. This is OK, but, How many uncoupled Cu(II) ions per 10 Cu(II) ions involved in the closed ring? A better description of the magnetic model is required.

Authors: According to referee’s suggestion, we have described the magnetic model of complex 2 in more detail. Please see the main text.

  1. It is always a good idea to include comparing the simulated and experimental powder diffraction patterns to ensure the integrity of the polycrystalline sample.

Authors: Thank you for your suggestion. Indeed, to ensure the integrity of the polycrystalline sample, comparing the simulated and experimental of XRD is a good method. Besides, elemental analyses can also indicate the purity of the sample to a certain extent. For compounds 1 and 2, the results of elemental analyses are very close to their theoretical values, indicating the integrity of two samples.

Reviewer 2 Report

The work entitled  “Two-Dimensional Nitronyl Nitroxide-Cu Networks Based on 2 Multi-dentate Nitronyl Nitroxides: Structures and Magnetic Properties” by Hongdao Li, Licun Li et al. reports the synthesis, structural analysis and magnetic properties for 2D heterospin 2p-3d coordination polymers based on copper ion and two two multidentate nitronyl nitroxide radicals.

The report is good, taking in account the complexity of the systems. However, we consider that the manuscript can be improved by answering to the following issues.

The affirmation “strong antiferromagnetic coupling and ferromagnetic interaction coexist in 1” from Abstract is not commented in the manuscript and can be valid as well for 2.

In the experimental section it specified that “Nitronyl nitroxide ligands bisNITPhPy and NIT-3Py-5-4Py were prepared according to literature method [51,52].” In the quoted references there are mentioned other compounds and a general synthetic method, so the authors are requested to add in detail the synthesis of both nitronyl nitroxide ligands used in synthesis of compound 1 and 2, if the synthesis was not published somewhere else.

Since there are different geometries for copper ions (distorted octahedral and distorted square pyramid or trigonal bipyramid for 1 and distorted octahedral for 2), it will be helpful to make for 1 a structural analysis including the five-coordinate geometry index. RPE spectra for both compounds will help the discussion.

It is important to present a scheme for magnetic exchange pathways for 1, similar to the Scheme 1 realized for 2.

The proposed spin polarization mechanism for the magnetic coupling mediated by NIT-3Py-5-4Py ligand presented in the Figure S1 is important for a deep understanding, but should be sustained by spin density calculation, otherwise remain just a supposition.

I considered the required revision as minor, evaluating positively the experimental part and the ensemble of the work and I draw the attention, in the same time that the outlined issues are important for a better understanding and improve the quality of the paper.

Author Response

Reviewer 2:

The work entitled  “Two-Dimensional Nitronyl Nitroxide-Cu Networks Based on 2 Multi-dentate Nitronyl Nitroxides: Structures and Magnetic Properties” by Hongdao Li, Licun Li et al. reports the synthesis, structural analysis and magnetic properties for 2D heterospin 2p-3d coordination polymers based on copper ion and two two multidentate nitronyl nitroxide radicals.

The report is good, taking in account the complexity of the systems. However, we consider that the manuscript can be improved by answering to the following issues.

  1. The affirmation “strong antiferromagnetic coupling and ferromagnetic interaction coexist in 1” from Abstract is not commented in the manuscript and can be valid as well for 2.

Authors: For complexes 1 and 2, both exist ferromagnetic because the NO groups are only coordinated to Cu(II) ions in the axial position. But for complex 2, there is no strong antiferromagnetic interaction due to the absent Cu-NO equatorial coordination bond. These have been illustrated in the paper.

  1. In the experimental section it specified that “Nitronyl nitroxide ligands bisNITPhPy and NIT-3Py-5-4Py were prepared according to literature method [51,52].” In the quoted references there are mentioned other compounds and a general synthetic method, so the authors are requested to add in detail the synthesis of both nitronyl nitroxide ligands used in synthesis of compound 1 and 2, if the synthesis was not published somewhere else.

Authors: These two radicals are not new and the cited references have been corrected.

  1. Since there are different geometries for copper ions (distorted octahedral and distorted square pyramid or trigonal bipyramid for 1 and distorted octahedral for 2), it will be helpful to make for 1 a structural analysis including the five-coordinate geometry index. RPE spectra for both compounds will help the discussion.

Authors: Thank you for your suggestion. According to referee’s suggestion, we performed shape analysis for Cu(II) ions in 1 and 2 (Table S3 and S4, ESI).

  1. It is important to present a scheme for magnetic exchange pathways for 1, similar to the Scheme 1 realized for 2.

Authors: Thank you for your suggestion. According to referee’s suggestion, we have indicated magnetic exchange pathways for compound 1 with a scheme, please see scheme 2.

  1. The proposed spin polarization mechanism for the magnetic coupling mediated by NIT-3Py-5-4Py ligand presented in the Figure S1 is important for a deep understanding, but should be sustained by spin density calculation, otherwise remain just a supposition.

Authors: Thank you for your suggestion. For CuII-NIT interaction through one pyridine ring, we have assessed qualitatively via spin polarization mechanism and analyzed via MAGPACK program. To some extent, the analysis of CuII-NIT interaction through one pyridine ring is sufficient. Regrettably, however, spin density calculation cannot be implemented due to the limited conditions at present.

Reviewer 3 Report

The manuscript describes synthesis, single crystal structure and magnetic properties of two new 2D Cu(II) complexes with nitronyl nitroxide radicals. The paper is clearly written, the methods used are appropriate for detail characterization of the compounds. The paper deserves to be published in Magnetochemistry after minor revision.

Remarks:

1) z is absent in the caption to Fig. 1 for (b) symmetry code.

2) ‘for Cu3’ (line 103) is repeated in the sentence and should be deleted.

3) High R1 value (0.0927) for the structure 2 and presence of large solvent accessible volume (184 A3 according to checkcif) means that solvent was not found but this is not described in the experimental section.

Author Response

Reviewer 3:

The manuscript describes synthesis, single crystal structure and magnetic properties of two new 2D Cu(II) complexes with nitronyl nitroxide radicals. The paper is clearly written, the methods used are appropriate for detail characterization of the compounds. The paper deserves to be published in Magnetochemistry after minor revision.

Remarks:

  1. z is absent in the caption to Fig. 1 for (b) symmetry code.

Authors: Thank you for your suggestion. According to referee’s suggestion, we have corrected the symmetry code, please see the main text.

  1. ‘for Cu3’ (line 103) is repeated in the sentence and should be deleted.

Authors: According to referee’s suggestion, we have deleted repeated words.

  1. High R1 value (0.0927) for the structure 2 and presence of large solvent accessible volume (184 A3 according to checkcif) means that solvent was not found but this is not described in the experimental section.

Authors: Thank you for your suggestion. We have added the description on large solvent accessible volume in the experimental section, please see.

Round 2

Reviewer 1 Report

Still a few spelling errors.

In Abstract line 12 "multi-denate" -> "multi-dentate"

Maybe Experimental portion should be called Experimental Section (page 8)

The rest is fine and now the work is aceptable after those minor changes.

Author Response

  1. Still a few spelling errors.

In Abstract line 12 "multi-denate" -> "multi-dentate"

Maybe Experimental portion should be called Experimental Section (page 8)

Authors: According to referee’s suggestion, we have corrected some spelling mistakes. And “Experimental portion” has corrected to “Experimental Section”
